

# Advances and complications of regenerative medicine in diabetes therapy

Olga Brovkina[1,*] and Erdem Dashinimaev[2,3,*]

[1] Federal Research Clinical Center for Specialized Types of Health Care and Medical Technologies of Federal Medical and Biology Agency, Moscow, Russia
[2] Koltzov Institute of Developmental Biology of Russian Academy of Sciences, Moscow, Russia
[3] Center for Precision Genome Editing and Genetic Technologies for Biomedicine, Pirogov Russian National Research Medical University, Moscow, Russia
[*] These authors contributed equally to this work.

## ABSTRACT

The rapid development of technologies in regenerative medicine indicates clearly that their common application is not a matter of if, but of when. However, the regeneration of beta-cells for diabetes patients remains a complex challenge due to the plurality of related problems. Indeed, the generation of beta-cells masses expressing marker genes is only a first step, with maintaining permanent insulin secretion, their protection from the immune system and avoiding pathological modifications in the genome being the necessary next developments. The prospects of regenerative medicine in diabetes therapy were promoted by the emergence of promising results with embryonic stem cells (ESCs). Their pluripotency and proliferation in an undifferentiated state during culture have ensured the success of ESCs in regenerative medicine. The discovery of induced pluripotent stem cells (iPSCs) derived from the patients' own mesenchymal cells has provided further hope for diabetes treatment. Nonetheless, the use of stem cells has significant limitations related to the pluripotent stage, such as the risk of development of teratomas. Thus, the direct conversion of mature cells into beta-cells could address this issue. Recent studies have shown the possibility of such transdifferentiation and have set trends for regeneration medicine, directed at minimizing genome modifications and invasive procedures. In this review, we will discuss the published results of beta-cell regeneration and the advantages and disadvantages illustrated by these experiments.

## INTRODUCTION

Diabetes mellitus is a widespread and socially significant disease that leads to a deterioration in the quality of life and life expectancy of patients. Patients with type I diabetes (T1D) are characterized by a deficiency of the pancreatic beta-cells mass, which can represent a loss of between 70–100% (*Lam & Cherney, 2018*). Traditionally, T1D can be separated from type II diabetes (T2D) by the factors triggering it, age of manifestation, and the useful

Corresponding author
Olga Brovkina, brov.olia@gmail.com

strategies for treatment. T2D is considered to develop due to insulin resistance rather than initial beta-cell loss. However, to compensate for the imbalance, the beta-cells produce more insulin but then, after years of hypersecretion, the pool of beta-cells is depleted by up to 65% in some cases. As a consequence, this group of T2D patients also needs exogenous insulin administration (*Butler et al., 2003*; *Eckel et al., 2011*).

However, exogenous control of insulin dosage cannot compensate for the sensitive adjustment normally made by beta-cells. This often leads to hyper- or hypoglycemia with corresponding complications increasing with the time. Hyperglycemia increases the development and progression of microvascular complications such as retinopathy, nephropathy, and neuropathy (*Gibbons & Freeman, 2020*; *Williams, Nawaz & Evans, 2020*). Furthermore, the most severe complications are those related to hypoglycemia in T1D patients, resulting in neurocognitive dysfunction (*Kodl & Seaquist, 2008*). Thus, recovery of the pool of beta-cells is an attractive strategy for treating patients with diabetes (Fig. 1). Currently, only surgical islet transplantation provides such an opportunity (*Shapiro, Pokrywczynska & Ricordi, 2017*; *Matsumoto & Shimoda, 2020*). Islet transplantation has been developed mainly for T1D patients and involves both transfusion into the portal vein and embolization within the liver, and allows the avoidance of complications related to whole organ transplantation. However, such an approach is complicated by the limited source of donor tissue available and by immune rejection, requiring immuno-suppression therapy (*Posselt et al., 2010*). Furthermore, there are no approved methods in regenerative medicine that can address the specific characteristics required of beta-cells.

Our current understanding of lesion mechanisms in beta-cells is based on rodent studies. Although it appears that human pancreas does show a corresponding lack of regeneration capacity under systemic disorders. Therefore the translation of these observation to human beta-cells should be handled with caution (*Zhou & Melton, 2018*). Indeed, a comprehensive analysis of a diverse array of animal models and human cell cultures might be fruitful in initiating new advances in regenerative medicine.

## SURVEY METHODOLOGY

The purpose of this article is to present a comprehensive review of the literature regarding recent methods of regenerative medicine in diabetes mellitus. To that aim, we performed PubMed searches for such keywords and terms as "beta-cells", "pluripotent stem cells", "3D culturing", "transdifferentiation", and "diabetes mellitus" among others. We then analyzed articles from Q1 journal ranking quartiles within the subdiscipline, using the SJR citations referenced. Although this is a comprehensive review, it is not exhaustive. It should be noted that we focused particularly on the limitations of the current methods, as solutions to these problems should help progress the development of regenerative medicine. However, it must be remembered, that the question regarding the problems underlying beta-cell preservation remains open and needs further investigation.

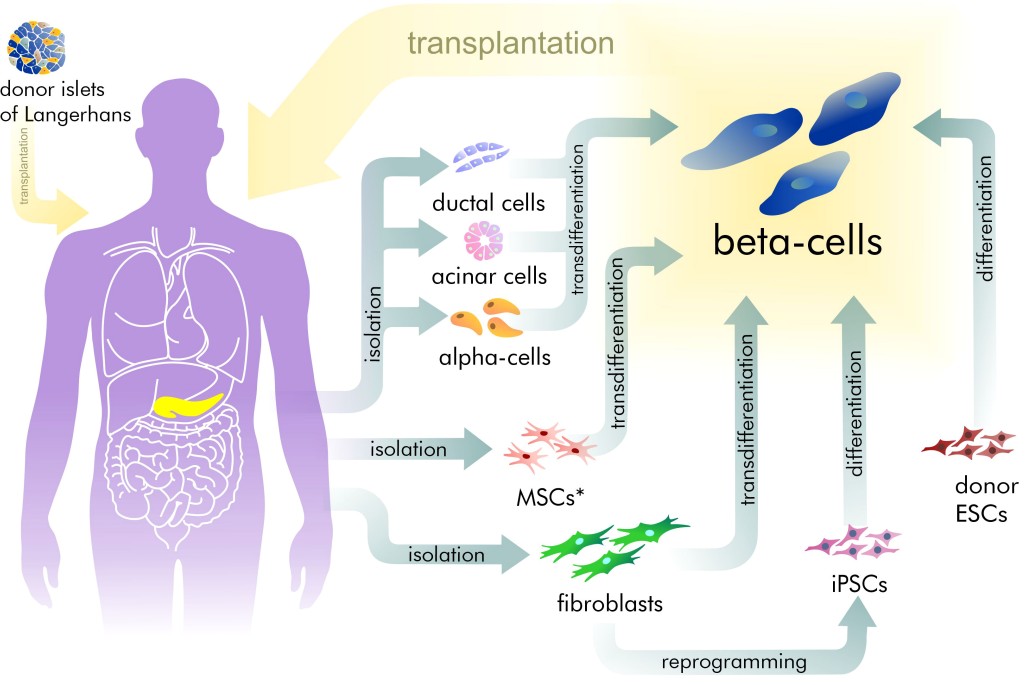

*Muscle-derived MSCs, adipose-derived MSCs, bone-marrow-derived MSCs

**Figure 1  The existing approaches for beta-cell recovery.**

## Characteristics of beta-cells

Beta-cells are one of the five cell types that form the islets of Langerhans, the hormone-producing structures of the pancreas. The islets are heterogeneously distributed in the pancreas and, according to recent studies, have a variety of content and functions. Patients with removal of the pancreatic tail showed increased glucose level indicators post-operatively than did patients with pancreatic head ectomy (*Menge et al., 2009*). Menge et al. showed, that the number of islets in the tail is greater than that in the head of the pancreas Such a difference can be explained by the distinct embryogeneses of the pancreas head and tail: while the head originates from the dorsal and ventral pancreatic bud, the body and tail are formed only from the ventral pancreatic bud (*Pandol, 2010*). That also influences the heterogeneity of the beta-cells, as has been confirmed by single-cell RNA-seq technologies. Recent studies have revealed two subpopulations of beta-cells with distinct molecular, physiological, and ultrastructural features (*Bader et al., 2016*; *Baron et al., 2016*). However, further investigations of their roles in diabetes are needed.

Beta-cell deficiency affects the ratios of other cell types in the islets of Langerhans. There are complex cell-to-cell interactions in the islets, resulting in a unique cytoarchitectural structure. Recent studies with beta-cell culturing in three-dimensional (3D) gels have shown the importance of such interactions (*Weber, Hayda & Anseth, 2008*; *Xu et al., 2020*). In these experiments the 3D gel contained extracellular matrix (ECM) proteins and directly influenced cytoskeletal organization, promoting beta-cell maturation and insulin secretion. As is known, diabetes patients exhibit dramatic changes not only in their hormone levels

but also in the structure and intrinsic cell communications within the pancreas. In this case, 3D hydrogel cultures are more promising for diabetes treatment than 2D ones.

In T1D the deficiency of beta-cells is a result of immune attack by T- lymphocytes. *Nakayama et al. (2007)* showed that in the NOD mouse model the insulin epitope is a prime target for effector T-lymphocytes: B:9-23 peptide promotes the development of insulin autoantibodies. But it is not only insulin production that is affected by this T-lymphocyte targeting; due to the effect of epitope spreading, the islet-specific glucose-6-phosphatase catalytic subunit–related protein (IGRP) is also targeted after insulin (*Prasad et al., 2012*). IGRP is a multicomponent integral membrane system that plays a crucial role in the terminal step of the gluconeogenic and glycogenolytic pathways, catalyzing the hydrolysis of glucose-6-phosphate. *Krishnamurthy et al. (2006)* demonstrated that the development of an autoimmune responses to multiple autoantigens increases the progression of T1D.

Herewith, histological analyzes show that the destruction of the beta-cells has a lobular character, and some of the islets show insulin staining (*Gianani et al., 2010*) This can be explained by the above-mentioned difference in the beta-cell populations.

## Beta-cell sources in vivo

To date, the existence of a definite native source for beta-cell regeneration remains in dispute (*Teta et al., 2007*). There are at least four suggested sources: beta-cells themselves, acinar cells, ductal cells, and alpha-cells.

Adult pancreatic beta-cells are formed mainly by self-duplication, this evidence having been demonstrated using a lineage tracing approach (*Dor et al., 2004*). Using a mouse model, Dor et al. showed that at least 85% of new beta-cells were formed from pre-existing beta-cells. This conclusion applied to both the normal adult pancreas and to pancreas subjected to partial ectomy.

However, special circumstances, e.g., pregnancy or diabetes, may activate other sources of beta-cells. Acinar, ductal and endocrine cells possess Pdx1/Ptf1-positive multipotent pancreatic progenitors during organogenesis. The endocrine specification of these progenitors is initiated by the Neurogenin3 (NEUROG3) and neurogenic differentiation 1 (NEUROD1) transcription factors (TFs), while the late stages of maturation of beta-cells is regulated by TFs including V-maf musculoaponeurotic fibrosarcoma oncogene homolog A (MAFA), V-maf musculoaponeurotic fibrosarcoma oncogene homolog B (MAFB), paired box gene 6 (PAX6), and estrogen-related receptor gamma (ESRRG) (*Zhu et al., 2017*). Thus, alterations in the ectopic expression of key transcription factors can change the fate of pancreatic cells (Table 1).

Embryonic ducts give rise to both differentiated endocrine and duct cells but once the pancreatic duct epithelium acquires a differentiated phenotype it does not contribute significantly to new beta-cells. Lineage-tracing studies by several research groups have shown that the ducts in adults do not appear to be a source of beta-cells (*Kopinke & Murtaugh, 2010*; *Kopp et al., 2011*). However, lineage-tracing analysis in studies by *Inada et al. (2008)* and *Zhang et al. (2016)* did demonstrate that such a possibility exists. Such discrepancies may be explained by heterogeneity in the duct cell populations and of the different gene regions that were used for tracing.

**Table 1   The key regulators of beta-cells.**

| Designation | Function | References |
|---|---|---|
| **Transcript factors** | | |
| Pdx1 | Essential in early and mature stages of pancreatic cell development. In early-stage regulates the formation of the pancreas from foregut endoderm. In the late stage transactivates the insulin gene | *Kaneto et al. (2007)* and *Zhu et al. (2017)* |
| Hb9 | Regulates development of the dorsal pancreas | *Arkhipova et al. (2012)* |
| Ngn3 | Critical regulator for pancreatic endocrine fates | *Akinci et al. (2012)*, p. 3 and *Zhu et al. (2017)* |
| NeuroD | Regulates pancreatic endocrine cell differentiation and insulin gene transcription | *Zhu et al. (2017)* |
| Mafa | Regulates glucose-responsive insulin secretion. This TF is expressed only in beta-cells | *Wang et al. (2007)*, *Nishimura, Bonner-Weir & Sharma (2009)* and *Zhu et al. (2017)* |
| Pax4 | Regulates of beta-cell specification | *Napolitano et al. (2015)* |
| Pax6 | Increases the insulin expression | *Arkhipova et al. (2012)* |
| Nkx6.1 | Guides endodermal progenitors toward beta-cell fate, must be generated before certain endocrine genes turn on | *Taylor, Liu & Sander (2013)* |
| Nkx2.2 | Critical regulator of pancreatic endocrine cell specification and differentiation | *Doyle & Sussel (2007)* |
| **Small molecules** | | |
| GNF-9228 | Selectively stimulates proliferation of beta-cells | *Pavathuparambil Abdul Manaph et al. (2019)* |
| 5′-azacytidine (5-AZA) | DNA methylation inhibitor; increases the reprogramming process 10-fold | *Hohmeier et al. (2020)* |
| Valproic acid (VPA) | Increases the reprogramming process 100-fold | *Hohmeier et al. (2020)* |
| Indolactam V (ILV) | Induces differentiation of hesc lines toward Pdx1-positive pancreatic progenitors | *Hohmeier et al. (2020)* |
| BRD7389 | Induces insulin protein expression in alpha-cells | *Hohmeier et al. (2020)* |
| Latrunculin A and B | Induce endocrine differentiation by the reorganization of the cytoskeleton | *Hogrebe et al. (2020)* |
| GABA (gamma-aminobutyric acid) | Induces alpha-to-beta-cell neogenesis; participates in maintaining the beta-cell mass and in protecting beta-cells from apoptosis in vitro | *Ben-Othman et al. (2017)* |
| **MicroRNAs** | | |
| miR-15a, miR-15b, miR-16 and miR-195 | Decrease the level of NGN3 gene expression | *Joglekar et al. (2007)* and *Wong et al. (2018)* |
| miR-7 | Inhibition of proliferation in Langerhans islets | *Wang et al. (2013)* and *Wong et al. (2018)* |
| **RNA-binding proteins** | | |
| Polypyrimidine tract-binding protein 1 (PTBP1) | Regulates the stability of insulin mRNA | *Magro & Solimena (2013)* |
| Human antigen D (HuD) | Binds to insulin mRNA and controls its translation | *Lee et al. (2012)* |
| Protein-disulfide isomerase (PDI) | Key regulator of glucose-stimulated insulin biosynthesis | *Kulkarni et al. (2011)* |

*Dirice et al. (2019)* showed a contribution of duct cells to the compensatory beta-cell pool by neogenesis during pregnancy, with increased insulin demand. Furthermore, using an injury model of the pancreas in mice, *Xu et al. (2008)* also revealed the neogenesis of

beta-cells from duct cells. The injury involved partial duct ligation with the restriction of digestive enzyme drainage from the exocrine pancreas into the duodenum. This ligation caused degeneration of the acinar cells and the extensive proliferation of duct cells. Xu et al. found that such ducts not only highly express NGN3, but also have the ability to form islets of Langerhans in NGN3-deficient embryos. This confirmed NGN3 as a key regulator both in beta-cell neogenesis and in their embryonic development.

Acinar cells are the most abundant type of exocrine pancreatic cells, producing enzymes for food digestion. They could also be a source of beta-cells, but attempts to prove this suggestion by lineage tracing methods in mouse models have been unsuccessful (*Desai et al., 2007*), indicating that acinar cells do not normally transdifferentiate into beta-cells. Nonetheless, by exhibiting significant transcriptional plasticity during culture, acinar cells remain an attractive source for in vitro beta-cell generation. Thus, *Pan et al. (2013)* showed the ability of acinar-to-beta-cell conversion, including an intermediate step with pancreatic multipotent progenitor cells. *Clayton et al. (2016)* performed direct reprogramming of acinar cells on a transgenic mouse by the virus-based induction of PDX1, MAFA, and NGN3. Another important output from their study concerns the impact of inflammation on the reprogramming result. Marked inflammation processes promote an acinar-to-ductal cell transition and inhibit the number of beta-cells.

Despite the fact, that alpha and beta-cells play antagonistic roles, alpha-cells seem to be the most promising source for beta-cell recovery, since they are the second-largest endocrine cell group in the pancreas and have tight connections with beta-cells. Several studies have demonstrated the possibility of alpha-to-beta transdifferentiation (*Thorel et al., 2010*; *Bru-Tari et al., 2019*). While the experiments include different diabetes modeling methods in mice, their results are united by the conclusion that the conversion can occur with immature alpha-cells. Only this limited population has the necessary plasticity for transition in the lack of insulin. To force this transition a Collombat group applied small molecules mimicking the effects of the ectopic expression of Pax4 and inhibiting Arx in alpha-cells (*Ben-Othman et al., 2017*). Small molecules refer to organic compounds with low molecular weight (<900 daltons), that may regulate biological process. A Collombat group identified that gamma-aminobutyric acid (GABA) induces alpha-to-beta transdifferentiation (Table 1). GABA is synthesized in beta-cells and has previously been described as a regulator of hormone secretion and as a signaling molecule in islet cell communication (*Franklin & Wollheim, 2004*). Notably, GABA-mediated transdifferentiation was limited by the number of alpha-cells, which decreased in line with the concomitant increase in numbers of beta-cells.

To sum up: pre-existing beta-cells appear to be the principal in vivo source of new beta-cells, while the contribution of other sources is less significant. However, several groups have demonstrated the feasibility of artificially inducing transdifferentiation of pancreatic non-beta-cells in vivo (*Zhou et al., 2008*; *Li et al., 2014*). Meanwhile, as the induced beta-cells are located in the native milieu, researchers have an opportunity for the direct comparison of endogenous and induced beta-cells. Despite promising results, there are some points that need further investigation. For instance, Zhou et al. observed a lack of organization of the induced beta-cells that can impair their function. Such result may

be explained by DNA methylation, which pre-existed in the regulatory elements of the non-beta-cells. Detailed epigenetic maps allow this problem to be overcome and a more targeted effect to be obtained (*Gifford & Meissner, 2012*). There are also many problems with integrating viral delivery, and these can be even more complicated in vivo, resulting in off-target effects.

## Beta-cell sources in vitro

*Ex-vivo* generation of beta-cells remains an attractive strategy in regeneration medicine, however, the differentiated cells normally have low proliferation activity. For these purposes, different agonists have been tested: nutrients, growth factors, intracellular signaling molecules, and small molecules (*Huang & Chang, 2014*). Currently, however, the proliferation of beta-cells in tissue culture results in a loss of the beta-cell phenotype, making it difficult to use them for diabetes therapy (*Efrat, 2008*). A proposed method of redifferentiation showed only low efficiency (*Kayali et al., 2007*).

To date, the most promising approaches for beta-cell generation include the differentiation of stem cells and the generation of beta-cells while bypassing pluripotency (Table 2).

1.ESC differentiation

The differentiation of ESCs into beta-cells in vitro was developed in the early 2000s (*Keller, 1995*). The Baetge group developed the first directed differentiation protocol and identified the main principles for stem cell differentiation into beta-cells (*D'Amour et al., 2006*). The first step in the differentiation of ESCs is a very critical stage in the formation of the definitive endoderm (DE) lineage (*Baetge, 2008*). This step is essential for the successful differentiation of the pancreatic lineage. The second step involves foregut endoderm formation and requires the addition of transforming growth factor-beta. Retinoic acid application is essential for the third step of the pancreas specification. Retinoic acid contributes to the efficient transition to the pancreatic lineage and prevents the differentiation of the pancreatic endoderm into endocrine cells. During the fourth step, foregut endoderm cells are recruited to the pancreatic and endocrine lineages. These cells have a high expression of PDX1 and transient expression of NGN3. During the fifth step the range of different hormones normally produced by endocrine cells start to be secreted. The ratio of beta-cells generated depends on the characteristics of the cell culture media and the success of the previous stages.

Each step requires strict monitoring of the expression of marker genes by immunohistochemical analysis, flow cytometry, and RT-PCR. Such an approach is essential for determining the homogeneity of the phenotypes and the efficiency of the multiple differentiation steps. However, even rigorous monitoring cannot guarantee the absence of teratoma formation from the undifferentiated ESCs. There is still a need for developing the standards needed in ESC therapy to provide safety (*Hentze et al., 2009*).

2. iPSC differentiation

The limited sources of ESCs and allograft rejection of transplanted ESCs have forced a search for alternative methods of beta-cell regeneration. The discovery of induced

**Table 2** The limitations of approaches for the generation of beta-cells.

| | Ex-vivo generation of beta-cells | ESCs differentiation in vitro | iPSCs differentiation in vitro | Non-beta pancreatic cells transdifferentiation in vivo | Non-beta pancreatic cells transdifferentiation in vitro | Fibroblasts transdifferentiation in vitro | References |
|---|---|---|---|---|---|---|---|
| Limited sources | Yes | Yes | No | Yes | Yes | No | *Huang & Chang (2014)* |
| Risk of teratoma development | No | Yes | Yes | No | No | No | *Hentze et al. (2009)* |
| Allograft rejection | No | Yes | No | No | No | No | *Hentze et al. (2009)* |
| Lack of organization into islets | No | Yes | Yes | Yes | Yes | Yes | *Zhou et al. (2008)* |
| Lack of reproducibility of the protocols | No | No | No | Yes | Yes | Yes | *Kim, Jeong & Choi (2020)* |
| Off-target effects after manipulation with genome | No | No | Yes/No[a] | Yes | Yes/No[a] | Yes/No[a] | *Clayton et al. (2016)* |
| The necessity of deep invasion for cell product preparation | Yes | No | No | No | Yes | No | *Trivedi et al. (2008)* and *Matsumoto & Shimoda (2020)* |
| The necessity of deep invasion for transplantation of final cell product | Yes | Yes | Yes | No | Yes | Yes | *Shapiro, Pokrywczynska & Ricordi (2017)* and *Matsumoto & Shimoda (2020)* |

**Notes.**
  [a]The presence of off-target effects will depend on the reprogramming methods (integrating or non-integrating).

pluripotent cells derived from human somatic cells has initiated a new chapter in regenerative medicine.

The use of iPSCs has allowed the generation of patient-specific beta-cells for autologous transplantation, with adult fibroblasts being frequently used for this process, representing an almost unlimited source for the generation of iPSCs.

Most of such iPSCs lines are made using retrovirus vectors that integrate the reprogramming factors involved in stem cell pluripotency. Such an approach is associated with the risk of tumorigenesis. Therefore the latest methods of iPSC generation involve non-integrating gene delivery, such as the Sendai virus, plasmid transfection, the piggyback transposon system, and minicircle vectors.

Recent studies have introduced a new strategy for the direct differentiation of iPSCs to pancreatic beta-cells (*Hogrebe et al., 2020*). The principle is based on cytoskeleton reorganization, whereby F-actin is depolymerized. It has been shown that latrunculin A and B guide stem cell differentiation towards a beta-cell fate, shortening the time for endocrine transcription factor expression and overcoming the requirement for 3D culture (Table 1).

These advances bring iPSC technology to the top of the most promising regeneration approaches. However, the risks found with teratoma formation from ESCs, for instance, also exist for iPSCs.

3. Mature cell transdifferentiation

The direct conversion of adult cells without the establishment of a pluripotency step would allow avoiding oncological complications. Early studies have revealed a capacity of pancreatic cells, such as ductal, acinar and non-beta endocrine cells to transdifferentiate into beta-cells by using a specific combination of multiple factors (PDX1, NGN3, and MAFA predominantly) (*Baeyens et al., 2005*; *Minami et al., 2005*).

While the pancreatic cells have increased capacity for being reprogrammed into beta-cells, there are still difficulties with culturing sufficient quantities of these cells. Besides, the pancreas is a retroperitoneal organ and is very difficult to assess for biopsies.

Thus, there is a need to search for alternative sources for transdifferentiation: ones that are both easy to culture and biopsy. In this case, human fibroblasts appear particulary promising. It has previously been shown that fibroblasts can be directly reprogrammed into neurons, hepatocytes, or cardiomyocytes (*Zhu, Wang & Ding, 2015*; *Babos & Ichida, 2015*; *Chen et al., 2017*). In these studies, the researchers applied a cell-activation and signaling-directed (CASD) strategy. In the first step of this approach, a key set of reprogramming factors (transcription factors or small molecules) turn the fibroblasts into a plastic and unstable state. During this step, the cells have a transient expression of pluripotency. However, several studies have shown, that such transient expression of the pluripotency genes does not necessarily mean a transient establishment of the pluripotency intrinsic for iPSCs (*Maza et al., 2015*; *De Los Angeles et al., 2015*). *Margariti et al. (2012)*, in their article, termed these "partial-iPS (PiPS) cells" and showed, that PiPS cells in this state did not form tumors in vivo. To provide an appropriate level of safety for the medical use of CASD technology, several researchers have suggested a CA step, using small molecules instead of transgenic reprogramming factors (Table 1) (*Hu et al., 2015*; *Li et al., 2015*). Furthermore,

such non-integrating methods make the CA step faster and more effective compared with manipulations of the genome. However, the low reproducibility of the protocols involving small molecules limits their application (*Kim, Jeong & Choi, 2020*).

Subsequent treatment with sequential combinations of growth factors and soluble lineage-specific signals proceed to the second step of the CASD approach. During the lineage-specific SD step the unstable intermediate population of cells generates progenitors of the desired cell line.

Currently, CASD technology has great potential in regenerative medicine and has raised hopes for its clinical application. However, further investigations are needed for comprehensive determination of the CA step and for obtaining homogeneity of cell conversion during the SD step.

4. Adult stem cell differentiation

One other promising source for beta-cell generation is adult stem cells (ASCs), and this has attracted close attention in recent years. ASCs, also called somatic stem cells, are located in specialized niches and serve for maintaining tissue homeostasis. Such niches with ASCs have been found in numerous tissues and organs, including the pancreas. However, these populations are rare and cannot provide complete regeneration of the beta-cell mass in patients with systemic endocrine disorder. Thus, various research groups have suggested proliferating ASCs in vitro (*Gurusamy et al., 2018*). *Mitutsova et al. (2017)* have shown, that non-adherent muscle-derived adult stem cells (MDSCs) have the potential to differentiate into beta-cells during in vitro culturing. In their experiment a multipotent stem cell population isolated from adult skeletal muscle was able to form islet-like cell clusters. Furthermore, adult MDSCs injected into hyperglycemic diabetic mice, successfully differentiated into beta-cells in vivo and contributed to reducing the glucose levels. Despite these promising results, the application of adult MDSCs is restricted by complications of the muscle biopsy, which requires several areas of expertise and optimal cryoprocessing of the fresh specimens. In the light of this, bone-marrow and adipose-derived stem cells (BMDSCs and ADSCs) are more readily available (*Trivedi et al., 2008*; *Phadnis et al., 2011*). BMDSCs contain two distinct populations: hematopoietic stem cells and mesenchymal stem cells, of which the mesenchymal stem cell population is more successful in adopting a pancreatic fate (*Carlsson et al., 2015*). *Zhao et al. (2007)* have identified a novel cell population of monocytes from adult human blood that can differentiate into efficient insulin-producing cells in vitro. These peripheral blood mononuclear cells express ESC transcription factors along with the hematopoietic markers (*Zhao et al., 2007*). The authors showed that, after 30 days of culturing, these mononuclear cells began to express the insulin gene transcription factors MafA and Nkx6.1 and to produce insulin.

ADSCs have certain advantages compared with BMDSCs (*Takahashi et al., 2019*). First the isolation of adipose-derived stem cells presents less harvesting difficulty (*Wang et al., 2018*). Second, ADSCs have been shown to have a higher differentiative potential, while BMDSCs tend toward osteogenesis and chondrogenesis (*Dmitrieva et al., 2012*). The third advantage involves the immunosuppressive capacity of ADSCs. It has been shown that ADSCs do not excite alloreactivity of incompatible lymphocytes, suppressing the mixed lymphocyte reaction by inducing the expansion of Tregs (*Puissant et al., 2005*).

Furthermore, ADSCs contribute to an increase in anti-inflammatory cytokines (*Kocan et al., 2017*). However, the increased expression of inflammatory cytokines and long-term fibrinolytic modifications in T2DM patients might impair the efficiency of therapy by adipose-derived stem cells (*Serena et al., 2016*; *Brovkina et al., 2019*).

Urine-derived stem cells (USCs) can also be applied as a potential stem cell source for diabetes therapy (*Zhao et al., 2018*). The advantages of USC use consists in their noninvasive isolation and expandable and simple methods of culturing. However, Zhao et al. noticed that USCs are useful only in compensating for minor apoptosis of beta-cells, but not in cases of extensive depletion as in T1D patients.

One of the major problem of the application of ASCs is their limited transdifferentiation efficiency, which affects from 5 to 25% of the cells (*Sharma & Rani, 2017*). Another problem is the low level of insulin secreted by differentiated ASCs and their limited lifespan without stemness. Nonetheless, despite these current limitations the ASC approach has undeniable advantages over using iPSCs since it presents a lower risk of generating teratomas and autoimmune responses in vivo.

## Methods for beta-cell preservation

The protection of beta-cells is no less important than their generation. This is an especially significant issue in the early phases of T1D or T2D when beta-cell mass is maximal. The common strategy for T2D treatment is targeting insulin-resistance and beta-cell dysfunction. Metformin and PPAR-gamma agonists are applied to decrease the insulin resistance, while sulphonylureas and glinides enhance secretion by the beta-cells. Some of the drugs effective in T2D treatment have also been considered for T1D patients. For example, verapamils, calcium channel blockers, and glucagon-like peptide-1 receptor agonists decrease beta-cell injury and exert beta cell-protective effects in experimental models of T1D (*Chen et al., 2008*; *Lotfy et al., 2014*). However, due to the immunogenic development of T1D, immunotherapy for beta-cells is more effective. Immunotherapy of T1D has two main approaches: the first is manipulation of the autoimmune response independently of beta-cell specificity, while the second uses beta-cell antigen-specific strategies. The limited specificity of the first approach can suppress acquired immunity. In this case, antibody-based therapy is more promising. However, current studies of anti-CD3 monoclonal antibodies show, that the protection of the residual beta-cell mass is short-term and can result in cytokine release and unwanted inflammation (*Chatenoud & Bluestone, 2007*, p. 3).

In regenerative medicine, the protection of generated beta-cells is also a paramount aim. However, first of all, we have to define what exactly should be protected—the beta-cell mass or its function. Beta-cells have different populations and some of them survive during immune attack (*Keenan et al., 2010*). Further investigation has revealed, that the surviving populations are of lower granularity cells with stem-like features and increased rates of proliferation, but they have a lower insulin content (*Rui et al., 2017*). *Talchai et al. (2012)* suggested these characteristics are due to beta-cell dedifferentiation. Thus, the reemergence of endocrine progenitor-like cells results in beta-cell dysfunction. In such cases the beta-cells maintain the expression of marker genes, like Pdx1 and MafA, but adopt

a dedifferentiated fate and do not release insulin. Recent work by Lee et al. confirms that dedifferentiated beta-cells exhibit reduced levels of beta-cell autoantigens and increased expression of immune inhibitory markers (*Lee et al., 2020*). Evidently, the preservation of beta-cell function and their mature populations should be a priority.

There are two main approaches that enable the protection of transplanted cells from the immune system without the need for systemic immunosuppression: islet encapsulation and "open" islet transplantation (*Korsgren, 2017*; *Peloso et al., 2018*; *Ernst et al., 2019*). The encapsulation strategy uses a biocompatible membrane that wraps the beta-cells. Such containers permit the diffusion of oxygen and nutrients while protecting against larger molecules, including antibodies. This strategy has several advantages. Since the mature beta-cells are more fragile, immature cells are preferable for transplantation. Encapsulation both isolates immature cells from stress conditions and protects them from the host immune system. A further advantage of encapsulation is the possibility of immediate removal of specific containers if tumorigenesis is detected.

Despite some promising results, current encapsulation technology has serious limitations. One of these is a difference in the kinetics of glucose and insulin diffusion compared with native islets (*Korsgren, 2017*). Insulin secretion in encapsulated beta-cells is delayed, leading to a risk of persistence of hypoglycemia. Another problem is of limited oxygen and nutrient delivery to the encapsulated cells due to deficiencies in the vascularization processes (*Rafael et al., 2000*). A third problem is the process of fibrosis that starts with recognition of nonspecific protein adsorption and leads to the formation of a fibrotic capsule around the implant. Nanotechnology provides opportunities to overcome or mitigate the aforementioned problems. A variety of nanoscale chemical, physical, and biological properties of the interfacing polymers has been shown to modulate the immune response and to enhance the vascularization processes. In this case, the composition of the polymer plays a significant role in the biocompatibility. Hydrogels and alginate matrix are commonly used in encapsulation technology. The modification of these polymers with certain compounds and proteins both decreases the magnitude of fibrosis and promotes vascularization. Thus, polyethylene glycol (PEG) with affinity peptides has been shown to significantly reduce the level of chemokines, thereby contributing to a reduction of inflammation (*Lin et al., 2010*). Hemoglobin-conjugated hydrogels improve the islet oxygenation, while crosslinked antioxidant enzymes can protect the hemoglobin from hypoxic and free radical stress (*Nadithe, Mishra & Bae, 2012*). Increasing vascularization is a challenging task, because this process is accompanied by inflammation, and this negatively influences islet survival. The current strategy for solving this problem includes modification of the surface roughness and porosity of the polymers, combined with anchoring vascular endothelial growth factor to their surfaces. Collagens, fibronectin, and laminin in the islet coat conduce the vascularity of implanted islets by mimicking the natural islet matrix (*Salvay et al., 2008*). *Vlahos, Cober & Sefton (2017)* engineered submillimeter collagen cylinders (modules) coated with endothelial cells, and these promoted islet vascularization and resulted in a proangiogenic M2-like macrophage response. Also, the larger pores were able to act as effective scaffolding for the vasculature. *Brauker et al. (1995)* demonstrated

that a 0.8–8-$\mu$m pore size allowed the penetration of host blood vessels into the device if the large-pore membrane was laminated to a smaller-pore inner membrane.

Finally, the layer-by-layer method allows the engineering of islet polymer films with versatile functions and optimal composition, thickness, and mechanical properties (*Wilson et al., 2011*).

In contrast to the encapsulation strategy "open" islet transplantation permits host interaction with the implanted material. The stem cell educator therapy proposed by Zhao et al. reverses autoimmunity and allows the preservation of transplanted islets (*Zhao et al., 2012*; *Zhao, 2012*). This therapy is based on cord blood-derived multipotent stem cells being co-cultured with the patient's lymphocytes in a closed-loop system. After this, the educated lymphocytes are separated and returned to the patient's circulation. The authors demonstrated that stem cell educator therapy increases in the number of Tregs and restores the cytokine balance (*Zhao et al., 2012*). Peripheral blood-derived insulin-producing cells are an alternative source that display the characteristics of islet beta-cell progenitors and therefore can also provide a potential resource for educator therapy (*Zhao et al., 2007*).

The application of stem cells may also play a part in the protection of the transplanted beta-cells. Such an impact is due to the ability to induce the proliferation of Tregs and the promotion of neovascularization (*Caballero et al., 2007*; *Chen et al., 2020*). Thus, a hybrid transplantation, which combined islet transplantation with mesenchymal stem cells, was shown to improve graft function (*Figliuzzi et al., 2009*). Furthermore, in vitro co-culturing of an islet graft with mesenchymal stem cells contributed to oxygenation and nutrient supply, as well as decreasing the levels of inflammatory cytokines (*Arzouni et al., 2017*). Therefore, co-culturing appears to increase both the survival of the beta-cells and islet quality.

Summing up, the development and combining of new technologies in regenerative medicine provides a means to overcome the very real complications of beta-cell recovery. However, despite the promising results, the effectiveness and reproducibility of the suggested solutions need further verification.

## CONCLUSIONS

These days regenerative medicine is advancing rapidly, breaking new ground. iPSC-technology is replacing ESCs, to provide an almost unlimited source for the generation of desired autologous cell lines. Currently, stem cell technologies are taking the lead in the regeneration of beta-cells, however, there are certain risks due to manipulations of the genome and the capacity of undifferentiated cells for forming teratomas. To overcome these limitations, the transient introduction of RNAs, recombinant proteins, or chemical mimics can substitute virus-mediated introduction. Meanwhile, the direct conversion of mature cells or the differentiation of adult stem cells can reduce the risk of teratoma development. Thus, the current trends of regenerative medicine are aiming towards safety and immune-compatibility.

However, despite the substantial success of regenerative medicine overall, there are still many whitespaces in diabetes treatment when using beta-cells. Without a primary

determination of the diversity and functions of different beta-cell populations, current technologies for beta-cell reproduction represent a colossus with feet of clay and will continue to have limitations.

Epitranscriptome modifications also require further investigations, especially for transdifferentiation approaches, since most of the studies suggest that the newly generated beta-cells retain the epigenomic landscape of the initial cell line.

Evidently, beta-cell preservation remains a stumbling block in diabetes treatment. Finding a solution to this problem represents the core task both for treating early stage patients and for patients with a total loss of beta-cells.

### Funding
This work was supported by a grant No 075-15-2019-1789 from the Ministry of Science and Higher Education of the Russian Federation allocated to the Center for Precision Genome Editing and Genetic Technologies for Biomedicine. The funders had no role in study design, data collection and analysis, decision to publish, or preparation of the manuscript.

### Grant Disclosures
The following grant information was disclosed by the authors:
Ministry of Science and Higher Education of the Russian Federation allocated to the Center for Precision Genome Editing and Genetic Technologies for Biomedicine: 075-15-2019-1789.

### Competing Interests
The authors declare there are no competing interests.

### Author Contributions
- Olga Brovkina conceived and designed the experiments, performed the experiments, analyzed the data, prepared figures and/or tables, authored or reviewed drafts of the paper, and approved the final draft.
- Erdem Dashinimaev conceived and designed the experiments, performed the experiments, authored or reviewed drafts of the paper, and approved the final draft.

### Data Availability
This work is a literature review without data.

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
