# Peer review of "Advances and complications of regenerative medicine in diabetes therapy"

_PeerJ, doi:10.7717/peerj.9746_

## Round 0.1 · original submission · Major Revisions

This review summarizes with a concisely but logically organized framework, the current advances and limitations in the use of regenerative therapy for the treatment of Diabetes.

However, it suffers from an excessive concision with insufficient citations of sources and lacks discussion of potential nanotechnological and stem cell biological developments to overcome the actual limitations in achieving the regeneration of beta cells to treat Diabetes (see for recent examples reviews by Ernst et al. 2019 Nanotechnology in cell replacement therapies for type 1 diabetes. Adv Drug Deliv Rev. 139:116-138 and by Takahashi et al. 2019. Regenerative and Transplantation Medicine: Cellular Therapy Using Adipose Tissue-Derived Mesenchymal Stromal Cells for Type 1 Diabetes Mellitus. J Clin Med. 2019 ;8(2).) In that regard, although not purporting to be exhaustive in a field with abundant reports over 30 years, the proposed review lacks citing few reports (some of which have been suggested by the two referees) and could be further improved in its readability by graphical illustration of the major approaches discussed and their complications.

Secondly, as also noted by both reviewers, in order to improve the readability of this manuscript, the English language and grammar requires corrections by thorough reading from a native-equivalent English speaker.

Reviewer 1 ·

Basic reporting

A major problem with the use of English. The review does not do a good job of summarizing this field.

Experimental design

lacking rigor

Validity of the findings

Many of the important advances in this field are not included.

Additional comments

This review is concerned with the very important goal of restoring the beta cell deficiency of diabetes but is disappointing in a number of ways. It was obvious from the beginning that the use of the English language was very poorly done, which makes the paper unacceptable. In addition, the authors have not done good job of summarizing the current state of this exciting field. Many important papers are not cited and a number of paper that are included are out of date.

A few specific problems include the following;

Line 51: cognitive dysfunction from hypoglycemia caused by insulin treatment is not a big issue for T2D but is for T1D.

Line 53: There are many good recent reviews of islet transplantation. Why select a review from 2001?

Line 77: This is not clear.

Line 94: This is also not clear. A “glucose-related phosphatase” is mentioned but not explained in enough detail.

Line 105: What is meant by “85% of presented beta-cells”. Again, problems with poor use of English.

Line 118: Actually, lineage tracing has show that duct cells can turn into beta cells. (Inada et al., 2008; Zhang et al., 2016)

Line 140: For alpha- to beta-cell conversion, papers from the groups of Herrera and Collombat should be cited (Ben-Othman et al., 2017; Thorel et al., 2010).

Line 152: what is meant by “epigenetic reprogramming”?
Line 218: There is not much excitement about fibroblasts being a very promising source for making new beta cells.
Line 219: The CASD strategy is not well described.


References:

Ben-Othman, N., Vieira, A., Courtney, M., Record, F., Gjernes, E., Avolio, F., Hadzic, B., Druelle, N., Napolitano, T., Navarro-Sanz, S., et al. (2017). Long-Term GABA Administration Induces Alpha Cell-Mediated Beta-like Cell Neogenesis. Cell 168, 73-85.e11.
Inada, A., Nienaber, C., Katsuta, H., Fujitani, Y., Levine, J., Morita, R., Sharma, A., and Bonner-Weir, S. (2008). Carbonic anhydrase II-positive pancreatic cells are progenitors for both endocrine and exocrine pancreas after birth. Proceedings of the National Academy of Sciences of the United States of America 105, 19915-19919.
Thorel, F., Nepote, V., Avril, I., Kohno, K., Desgraz, R., Chera, S., and Herrera, P.L. (2010). Conversion of adult pancreatic alpha-cells to beta-cells after extreme beta-cell loss. Nature 464, 1149-1154.
Zhang, M., Lin, Q., Qi, T., Wang, T., Chen, C.C., Riggs, A.D., and Zeng, D. (2016). Growth factors and medium hyperglycemia induce Sox9+ ductal cell differentiation into beta cells in mice with reversal of diabetes. Proceedings of the National Academy of Sciences of the United States of America 113, 650-655.

Reviewer 2 ·

Basic reporting

In this review, Brovkina and Dashinimaev have chosen to address an interesting and important aspect concerning the potential use of stem cells and regenerative medicine in the treatment of diabetes. Indeed, the field of regenerative medicine and its application to diabetes, particularly beta-cell replacement and increasing beta-cell mass are areas currently expanding very rapidly making such a review timely and relevant. Unfortunately, in its present form, this review falls short of the author’s ambitious goals.

1. There is clearly a need for the authors to improve the English and preferably have the manuscript read and corrected by a native English speaker. There are many examples throughout where the text is cumbersome and in some places, it is difficult to understand the authors meaning at all. Examples in the abstract where the text is difficult to understand are lines: 18-19, … ensures that future application in the clinic is not a matter of if, but when. lines 20-21 “Indeed a generation of beta-cells mass with the expression of marker genes …. “ would read more easily as … “Indeed, the generation of functional beta-cell mass …”

Experimental design

2. The scope of the review is essentially correct and the review is organized logically and coherently. In overall depth, the review falls short in some aspects, for example, naturally occurring adult stem cells (somatic stem cells) from skeletal muscle (1), smooth muscle urinary tract (2) and intestine, blood, skin, brain, adipose (3) and heart tissue all well-reviewed in (4) have all shown promise in treating beta-cell deficiencies. It would seem appropriate to acknowledge this in new subsection 4 after Mature cell transdifferentiation.
While recognizing the size of the field, the review by Zhou and Melton 2018 (5) should be included in section 2.

1. Mitutsova V et al. Stem Cell Res Ther. (2017) 8:86
2. Zhao T et al., J Mol Histol. (2018) 49:419-428
3.Wang M. et al. Mol Ther. (2018) 26:1921-1930
4. Gurusamy N et al. Prog Mol Biol Transl Sci. (2018). 160: 1-22.
5, Zhou Q and Melton DA, (2018) Nature 557:351-358.

Validity of the findings

3. The conclusions are well stated and in agreement with the objectives set out in the introduction. However, in addition to emphasizing the prospective positive contribution of stem cells to this field, the conclusions should reiterate potential risks inherent in the use of either ESC or iPSC which are the major constraints to advancing this technology to the clinic.

---

## Round 0.2 · Minor Revisions

The revised manuscript by Brovkina and Dashinimaev has been improved both in its coverage of the reviewed literature and in the editing of the English language.

As a further improvement to a better valorisation of this review, it is suggested that Table 2 on “The limitations of approaches for the generation of beta-cells” be complemented by the addition of relevant references for each approach described in the top row of Table 2.

Reviewer 2 ·

Basic reporting

The authors have significantly improved the legibility of this review and it now reads easily. The article cites a broad range of literature in the field and the background and context of the review are clearly developed. The review is within the scope of the journal and the originality stems from the comprehensive nature in which the field has been analyzed.

Experimental design

The field has been reviewed rigorously and the methodologies are outlined clearly and provide an unbiased and comprehensive coverage of the subject. The review is logically organized.

Validity of the findings

The review clearly illustrates the current limitations on the use of stem cell therapies for the treatment of diabetes mellitus.

Additional comments

There are still a few errors in the English, which I outline here.
In the abstract line 20 diabetes --> diabetic patients.
Same line ... to the plurality... --> to a plurality.
Line 21 beta-cells masses --> beta-cell mass
Line 65 disorders lacks the full stop.
Line 78 should progress --> should help progress
Line 115 replace evidence --> show
Line 252 with biopsies --> for biopsies.
Line 408 of an islets graft --> of an islet graft.

---

## Round 0.3 · accepted · Accept

The last minor corrections and additions to your manuscript have been made and it is now significantly improved and acceptable for publication in PeerJ.